# Nature Photographs as Complementary Care in Chemotherapy: A Randomized Clinical Trial

**DOI:** 10.3390/ijerph20166555

**Published:** 2023-08-10

**Authors:** Giulia Catissi, Leticia Bernardes de Oliveira, Elivane da Silva Victor, Roberta Maria Savieto, Gustavo Benvenutti Borba, Erika Hingst-Zaher, Luciano Moreira Lima, Sabrina Bortolossi Bomfim, Eliseth Ribeiro Leão

**Affiliations:** 1Albert Einstein Israeli Faculty of Health Sciences, Hospital Israelita Albert Einstein, São Paulo 05651-901, Brazil; giulia.catissidelima@einstein.br; 2Pediatric and Neonatology Unit, Hospital Israelita Albert Einstein, São Paulo 05651-901, Brazil; leticia.bo@einstein.br; 3Albert Einstein Education and Research Center, Hospital Israelita Albert Einstein, São Paulo 05651-901, Brazil; elivane.victor@einstein.br (E.d.S.V.); roberta.savieto@einstein.br (R.M.S.); sabrina.bomfim@einstein.br (S.B.B.); 4Department of Electronics-DAELN, Graduate School on Biomedical Engineering—PPGEB, Federal University of Technology-Paraná—UTFPR, Curitiba 80230-901, Brazil; gustavobborba@utfpr.edu.br; 5Instituto Butantan, São Paulo 05503-900, Brazil; erika.zaher@butantan.gov.br (E.H.-Z.); calyptura@gmail.com (L.M.L.)

**Keywords:** nature-based therapy, complementary therapies, emotions, affect, signs and symptoms, nature, photography, integrative oncology

## Abstract

The incidence of cancer cases is increasing worldwide, and chemotherapy is often necessary as part of the treatment for many of these cases. Nature-based interventions have been shown to offer potential benefits for human well-being. Objective: This study aims to investigate the outcome of nature images on clinical symptom management related to chemotherapy. Methods: A randomized clinical trial was conducted in an outpatient cancer unit of a private hospital in Brazil, with 173 participants over the age of 18 who were undergoing chemotherapy and had signed an informed consent form. The intervention consisted of the presentation of a 12-min video featuring nature images categorized under the themes of Tranquility, Beauty, Emotions Up, or Miscellany. Images were sourced from the e-Nature Positive Emotions Photography Database (e-NatPOEM), a publicly available collection of affectively rated images. Sociodemographic and clinical data, as well as the participants’ connection to nature, were investigated. The Positive Affect/Negative Affect Scale (PANAS) and the Edmonton Symptom Assessment System (ESAS) were applied pre- and post-intervention. Results: Data showed very strong evidence of a reduction in negative affect for the intervention group (*p* < 0.001) and moderate evidence for the control group (*p* = 0.034). There was also a significant reduction in the intervention group for pain (*p* < 0.001), tiredness (*p* = 0.002), sadness (*p* < 0.001), anxiety (*p* < 0.001), and appetite (*p* = 0.001). The Beauty video had the best performance, while the Tranquility video showed no significant improvement in any of the symptoms evaluated. These findings suggest that images of nature may be a valuable tool to help control clinical and psychological symptoms in cancer patients undergoing chemotherapy.

## 1. Introduction

An estimated 19.3 million new cancer cases occurred worldwide in 2020, according to data from the World Health Organization and the American Cancer Society [1]. Cancer is a global public health problem with a significant clinical, social, and economic impact, resulting in high disability-adjusted life years compared to other diseases. The incidence, prevalence, and mortality of cancer are expected to increase over the next 40 years [1,2]. In 2020, a total of 18 million new cases were diagnosed, with lung, breast, and prostate cancer being the most common types. However, cancer rates vary widely among and within countries, depending on economic development and associated social and lifestyle factors [3]. While cancer treatments have made significant advancements in recent decades, including immunotherapy, precision medicine, chemotherapy, surgery, and radiotherapy, there is still much work to be done to improve outcomes for patients globally [4,5].

Cancer patients often experience negative emotions at diagnosis and throughout treatment [6]. In clinical practice, patients express fears, sadness, anguish, and anxiety due to the uncertainty of their prognosis [7]. During chemotherapy, patients require support to manage pain, weakness/fatigue, ominous feelings, mood changes, fear of cancer spreading, and death. These physiological and psychological symptoms can interfere with treatment adherence, underscoring the need for effective symptom management. In addition to medication administration, non-pharmacological interventions and light care technologies should be part of the therapeutic arsenal [8]. The hospital environment where patients receive treatment is also important.

Since Florence Nightingale, there has been a strong emphasis on creating restorative environments in hospitals. Florence recognized the importance of patients being able to see the sky and sunlight through a window, even if nothing else was possible, stating that such measures were of prime importance for patient recovery [9].

A window with a view of a natural landscape has been identified as a potential adjunct in the recovery of surgical patients, resulting in reduced analgesic consumption and shorter hospital stays [10]. However, most hospital constructions in urban contexts do not always provide such a view. This limitation has driven researchers’ interest in developing other forms of contact with natural elements that can bring some degree of well-being to patients during hospitalization and treatment, including the use of photographs and videos as a means of indirect interaction with nature [11].

The growing literature on nature-based interventions in various health settings suggests their potential to alleviate disease-related tension and positively impact patients. However, little is known about the role of human-nature interactions in patients’ health and illness experiences and whether nature contributes to recovery, health, and well-being from their perspective [12]. Researchers that use landscape photos and natural environment films as mediating elements to create a restorative environment indicate the potential for psychophysiological alterations and stress relief [13].

A review study has shown that cancer patients appreciate direct contact with nature and benefit from the opportunities of this connection, experiencing consequent tension relief (related to the diagnosis) and feeling aesthetically enriched by nature through the appreciation of its beauty, peace, tranquility, and solitude [12]. Although images of nature were not included in this review, it strengthens the hypothesis that contemplating them can be a valuable resource for patients undergoing chemotherapy.

In this study, we evaluated the therapeutic potential of viewing images of nature on the cancer patients’ positive/negative affect and clinical symptoms related to chemotherapy. For the assessments, we employed the Positive Affect/Negative Affect Scale (PANAS) [14] and the Edmonton Symptom Assessment System (ESAS) [15]. The nature images were obtained from the e-Nature Positive Emotions Photography Database (e-NatPOEM) [16]. The study included a total of 173 participants.

## 2. Materials and Methods

### 2.1. Study Design and Population

This randomized clinical trial was conducted in an outpatient care unit for cancer patients at an extra-large private general hospital located in the southern part of Sao Paulo, Brazil. All participants provided written informed consent. The study was approved by the Institutional Research Ethics Review Board in accordance with the Declaration of Helsinki and was registered with ClinicalTrials.gov Identifier: NCT03518255 (accessed on 3 May 2023).

The inclusion criteria comprised patients of both sexes, over 18 years old, with preserved clinical conditions and communication functions that allowed them to participate in the study. They were undergoing infusion chemotherapy for any type of cancer. Exclusion criteria comprised patients who received immunotherapy infusion, individuals with visual impairment, and those who experienced any side effects or deterioration in clinical condition (e.g., excessive sleepiness, fatigue, cognitive limitations) during the interview/intervention period.

Since antiemetic medications and certain pain control drugs can affect patients’ psychological state, the control and intervention groups received a similar drug protocol. The institution conducting this study is certified by the American Society of Clinical Oncology (ASCO) [17] and strictly adheres to well-defined protocols that undergo regular review by the clinical staff. Regarding clinical protocol for all participants, each patient underwent evaluation, and antiemetic prophylaxis was prescribed before initiating chemotherapy sessions. The selection of antiemetics was based on the emetic risk associated with the chemotherapy drugs, as well as patient-specific factors. High therapeutic index antiemetics included 1st generation 5-HT3 antagonists (such as Ondansetron and Granisetron), 2nd generation antagonists (such as palonosetron), corticosteroids, and neurokinin-1 receptor (NK1-R) antagonists (such as Aprepitant and Fosaprepitant). In cases where corticosteroids were contraindicated, low therapeutic index agents like metoclopramide, phenothiazines, and benzodiazepines were utilized.

### 2.2. Randomization and Sample Size

The potential participants were selected from a weekly list of patients scheduled for chemotherapy sessions in the upcoming week. Invitation e-mails were sent to the patients, and those who provided informed consent were included in the screening process.

From 22 August 2018 to 13 March 2020, 265 patients underwent screening for eligibility in the study. Among them, 35 were ineligible as they did not meet the inclusion criteria (incomplete chemotherapy sessions, undergoing immunotherapy, non-oncologic diagnosis, or inadequate clinical conditions). Additionally, 54 patients declined to participate, and one patient was isolated due to contact precaution procedures. The remaining 175 patients were randomly assigned to one of five groups: the Control group, intervention Beauty group, intervention Emotions Up group, intervention Miscellany group, and intervention Tranquility group. The randomization process was conducted using the Randomizer [18] software. Following randomization, two patients (one from the Control group and one from the Tranquility group) withdrew from the study due to symptoms experienced during the chemotherapy session, such as drowsiness and nausea. As a result, the study included a total of 173 participants. The complete flow diagram of the study participants is provided in Figure 1.

We hypothesized that the intervention group, exposed to images of nature, will experience a clinically significant improvement in positive affect and an 80% reduction in negative affect and chemotherapy-related symptoms/adverse events, compared to the control group. To detect a significant difference with 80% power at a 0.05 level of significance (two-sided), we calculated a sample size of at least 30 patients per group.

### 2.3. Data Sources and Measures

Participants completed a questionnaire that assessed their sociodemographic and clinical variables, including their sex, age, location (state, city, country), marital status, education, profession, urban/rural background, intentional relationship with nature, diagnosis, chemotherapy session, time of diagnosis, and their estimated treatment time for sample characterization.

To evaluate the affective aspect of the participants’ relationship with the environment, the Connectedness to Nature Scale (CNS) was used. The CNS consists of 14 items, such as “I think of nature as a community of which I am a part” and “My personal well-being is independent of the well-being of nature”. Participants rated their agreement on a scale ranging from 1 (I totally disagree) to 5 (I totally agree). This instrument was translated and validated in Brazil and has demonstrated to be a psychometrically adequate measure to assess the general factor of connection with nature in our environment [19].

The Positive Affect/Negative Affect Scale (PANAS) and the Edmonton Symptom Assessment System (ESAS) were administered pre-and post-intervention to all intervention and control groups to assess the primary outcomes. The PANAS consists of 20 items that describe feelings and emotions and assess a person’s positive and negative traits using a 5-point scale (1 = “very slightly or not at all”; 5 = “extremely”). Higher scores on Positive Affect indicate greater intensity of positive emotions, and higher scores on Negative Affect indicate greater intensity of negative emotions [14]. This questionnaire was validated in Brazil and presents adequate psychometric properties [20].

The Edmonton Symptom Assessment System (ESAS) addresses ten common symptoms: pain, tiredness, nausea, depression, anxiety, drowsiness, appetite, well-being, sleep, and shortness of breath. The severity at the time of assessment of each symptom is rated from 0 to 10 on a numerical scale; with 0 meaning that the symptom is absent and 10 that it is the worst possible severity. For all items, the higher the score, the worse the symptoms [15]. ESAS is considered a reliable and valid instrument for use in Brazil to assess symptoms in advanced cancer patients [19].

Assessments from the intervention and control groups were collected as follows: participants filled out pre-questionnaires before the session and post-questionnaires after a 20-min interval.

### 2.4. Intervention

The control group did not receive any media or other material. In the intervention group, following the initiation of the chemotherapy infusion procedure, a research team member provided a tablet (9.7 inches, 2048 × 1536 pixels) exclusively designated for the study and with the other functions restricted. The participant watched a 12-min video on the tablet in a private and comfortable environment, primarily within the room/bed where chemotherapy was administered.

The video watched by the participant was randomly selected from four different options: Beauty, Emotions Up, Tranquility, and Miscellany. Each video consisted of 60 images pre-selected from the e-NatPOEM database [16], which were randomly presented in a slideshow format. Randomizations were performed using Randomizer [18] software. The e-NatPOEM provides 400+ high-quality nature images, each with corresponding valence and arousal ratings, and semantical classification into the following categories—beauty, peace/tranquility, positive states, miscellaneous, and negative states. The validation procedures for this database are described in detail in [16].

The 240 distinct images (60 for each video) were pre-selected from the e-NatPOEM according to the criteria below. The corresponding images can be found in Appendix A.

Beauty: related to awe/fascination. Includes white and colorful birds, insects, flowers, and landscapes. Valence range: 6.9–7.8; Arousal range: 3.1–4.2.Emotions Up: related to joy and liveliness. Includes water, pale birds, colorful birds, sky, insects, and sea. Valence range: 5.8–8.0; Arousal range: 3.3–5.6.Tranquility: related to peace and calm. Includes water, sky, and sea. Valence range: 6.6–8.0; Arousal range: 2.8–4.2.Miscellany: includes water, trees, white birds, colorful birds, sky, flowers, insects, and landscapes. Valence range: 5.3–7.5; Arousal range: 3.0–6.1.

It is worth emphasizing that both groups received identical standards of care and treatment throughout the study, adhering to the criteria set by the Joint Commission International [21] and Magnet Recognition Program [22].

### 2.5. Statistical Analysis

The calculation of the Connectedness to Nature Scale followed the recommended procedure outlined in a previous study that evaluated the psychometric properties of the Brazilian version [19]. This involved summing the scores of the first thirteen items and reversing the ratings for items 4 and 7. The total score ranged from 13 to 65, with higher scores indicating a stronger connection to nature.

Variables were described in total and by groups using absolute frequencies and percentages for qualitative variables, and means, standard deviations, and minimum and maximum values for quantitative variables. Changes in symptoms on the Edmonton Scale were described by individual profile graphs.

To compare between groups and timepoints, we used the generalized estimation equation models [23], which consider the negative binomial distribution for Edmonton symptoms and the gamma distribution for positive and negative aspects. The link function was logarithmic, and the results of the models were presented as estimated mean values, with 95% confidence intervals (95% CI), and *p*-values for moment or group comparisons, which were corrected using the sequential Bonferroni method.

To evaluate the relationship between the Connectedness to Nature Scale and changes in positive and negative affect over timepoints, we used generalized linear models with a quasi-likelihood distribution, in order to adequately fit positive and negative changes, without normal distribution.

## 3. Results

### 3.1. Participant Characterization

We analyzed data from 173 individuals: 32 in the Control group (18.5%), 33 in the Miscellaneous group (19.1%), 39 in the Tranquility group (22.5%), 33 in the Emotions Up group (19.1%), and 36 in the Beauty group (20.8%). Overall, 81.5% of the sample received some intervention. Most participants were from Brazil (93.6%), born and currently living in urban areas (94.2% and 84.4%, respectively). Participants’ characteristics are presented in Table 1.

The mean age was 57 years, ranging from 22 to 90 years. The distribution by sex was 56.6% female, and the most frequent marital status was married. Regarding education level, 48.6% had completed undergraduate studies, and 33.5% had completed graduate studies.

Intentional contact with nature was reported as daily by 41% of participants, with 6.4% reporting just once a year. The Nature Connection Scale had a concentration in higher scale values, indicating a general high nature connection among participants.

A wide range of diagnoses was observed, with the most frequent being digestive system cancer (27.8%), followed by hematologic (21.0%) and reproductive system (20.9%) cancer. The current chemotherapy session number was an average of ten. The chemotherapy session at the time of the study was reported as the first for 41.0% of the participants, while the others were on average in their tenth session.

Regarding the time since diagnosis, 25.4% reported knowing about the disease between 3 and 6 months ago, as well as between 1 and 5 years ago. The estimated treatment time exceeded three months for more than 90% of the participants (Table 2).

### 3.2. Positive and Negative Affect

We compared the intervention groups and control group regarding negative affect at pre- and post-moments, and identified a significant reduction in all groups, including the control group (Table 3). Feeling afraid, nervous, distressed, restless, and upset were the negative affects that were most altered, with an improvement ranging from 16 to 30 points between pre- and post-moments.

The largest reduction, as indicated by estimated mean values and graphs, was observed in the Miscellany group (Figure 2). When considering positive affect, there was no evidence of significant changes between pre- and post-intervention, nor were there any differences between groups.

### 3.3. Edmonton Symptom Assessment System (ESAS)

We compared the intervention and control groups regarding symptoms at pre- and post-intervention moments and identified a reduction in the intensity of most symptoms (Table 4).

When considering pain, fatigue, sadness, anxiety, and appetite, there is evidence of significant improvement in symptoms across the entire intervention group. When evaluating the groups separately by an intervention video (Table 5 and Table 6), we observed an improvement in pain, fatigue, sadness, anxiety, appetite, well-being, and shortness of breath, which varied according to the content of each video (Table 7). For nausea, drowsiness, and sleep, we did not obtain evidence of a significant reduction between the moments for any of the groups.

## 4. Discussion

The present study assessed the therapeutic potential of nature image viewing in terms of its impact on cancer patients’ positive and negative affect, as well as the clinical symptoms associated with chemotherapy.

Anxiety is one of the most prevalent mood states during cancer treatment, related to the diagnosis and treatment itself, fear of metastasis, the unpredictability of physical suffering, and the future. In comparison to the general population, cancer patients exhibit lower levels of positive emotions, as evaluated by PANAS. However, there is no significant difference in the prevalence of negative emotions, indicating similarity in negative emotional experiences [24]. The psychological distress of cancer patients seems to be mainly caused by low levels of positive affect. The prevalence of emotional distress in cancer patients varies from 35% to 55% [25], highlighting the need for interventions that promote the enhancement or maintenance of positive affect during treatment.

The PANAS scores for positive affect, as evaluated in our study, were approximately 25% higher both before and after the intervention, compared to patients with advanced cancer, or even the general population [25]. This may partly explain the absence of significant changes in positive affect. Other factors may have influenced this, such as confidence in the healthcare team and the institution where chemotherapy was performed, which is highly rated in international rankings in terms of quality of care. The fact that half of the sample was undergoing the beginning of treatment may have resulted in greater optimism. The PANAS scores for negative affect, as assessed in our study, were 17.15 for the intervention group and 16.30 for the control group in pre-intervention, which is similar to the scores observed in patients with advanced cancer (17.6) [25]. Although positive affect plays an important role in psychological distress, it has been observed that interventions with this population have a greater influence on reducing negative affect than on increasing positive affect, without, however, failing to contribute to subjective well-being and the reduction of psychological distress [25].

All intervention groups experienced a significant reduction in negative affect. The main negative affect states that showed significant decreases included feelings of fear, nervousness, distress, restlessness, and upset. The control group also showed moderate evidence of reduced negative affect, including anxiety evaluated by the ESAS. This effect can however be expected at the end of a chemotherapy session that has transpired without complications for the study participants.

Regarding video content, it is worth noting that the largest reduction in negative affect occurred for the miscellaneous video.

The video contents also influenced clinical symptoms in a differentiated manner. The best-performing was the Beauty video, which reduced pain most significantly but also improved sadness and anxiety, and improved well-being and shortness of breath. Beauty is a psychological sensation perceived as a sensory impression; this sensation is pleasant and positive, which can alter the perceptual focus of pain and reduce stress and anxiety by modulating attention as well as higher-order cognitive and emotional processes [26].

The Emotions Up video altered the perception of pain, fatigue, anxiety, and appetite. Surprisingly, the Tranquility video, consisting mainly of images of the sky and sea, with low scores of arousal, had no impact on the participants (note that even the control group showed a reduction in anxiety). This may have occurred because although the images individually evoked feelings of peace and tranquility, but when proposed in the same video, they may have become monotonous in the face of the same predominance of color and theme.

Another possibility is that, similarly to what happens with experiments that use music to induce emotions, the ISO (from the Greek “isos”, meaning equal) principle may also affect visual stimuli. The ISO principle states the importance of inducing emotional changes gradually. This involves starting with music related to the subject’s initial emotion and smoothly transitioning to subsequent emotion stimuli, ultimately achieving the desired emotion. This implies that abrupt emotional shifts induced by external stimuli are not readily accepted by individuals. Therefore, it is important to modulate the induced emotions [27]. The feeling of tranquility imposed by images without any gradation may have sounded almost like a challenge to the anxiety related to the chemotherapy procedure and instead of attenuating it, may have reinforced its perception, which was reflected in the absence of any influence. This leads us to recommend attention even to images that promote positive emotions, as observed in the general population, as they may have variability depending on the audience and context.

The Miscellany video reduced fatigue, sadness, and anxiety, and presented the best performance in reducing the negative affect as assessed by PANAS. The miscellany video was compounded by different nature categories, which may confer greater mental fluctuation, without a more specific emotion or experience. Combining these diverse images may have exerted a more realistic and comprehensive result, resembling the broader experience of nature rather than the repetition of a few specific natural elements. The inclusion of elements such as working ants, butterflies, flower-filled paths, and landscapes could promote a sense of proximity and familiarity, while the vibrant colors used in the images might have influenced the perception of fatigue, making them more visually appealing. Additionally, these images presented together also gathered a variety of attributes, such as light, heat, attention, alertness, strength, curiosity, and lightness, among many others mentioned in [16].

In the Emotions Up video, the images also comprised a thematic diversity, which may have favored distraction through the stimulation of visual neurons. Among the mechanisms related to pain relief, distraction and alteration of perceptual focus have been emphasized. These mechanisms can result in the release of endorphins, resulting from both pleasure sensations and positive memories. The brain processing of images permeates all these mechanisms. Positive emotions can help maintain a positive psychological state and neutralize negative situations, including painful experiences [28]. Visual stimuli have been associated with the reduction of pain and anxiety in other procedures in a hospital setting, such as for colonoscopies [29].

Reduction of the negative affect observed in the control group may have been influenced by the researchers’ contact with the patients during data collection, potentially impacting the patients’ relaxation levels. Social contact provides a sense of connection and support, alleviating loneliness and fear. Additionally, social interactions serve as a distraction from the stress and discomfort associated with treatment, promoting a more positive emotional state. Social contact also provides a sense of normalcy and helps patients maintain a connection with the outside world, offering emotional comfort, and encouragement, and enhancing resilience [30,31]. An uneventful chemotherapy session (since patients have negative expectations that it will occur) may also result in an expected reduction in anxiety.

Fatigue is another symptom that is quite prevalent among patients receiving venous or oral chemotherapy. Researchers have shown that viewing nature photos for just six minutes provides a quick and temporary boost, similar to caffeine, improving executive attention in young and old adults, with a restorative influence in older adults who tend to tire more easily and experience more fatigue [26].

A comparative study between forest photos presented on a conventional screen and urban photos presented in 360° format demonstrated that forest photos, regardless of the presentation format, are more effective in preventing mood disturbances compared to urban immersive 360° photos [32].

Photographs, art, and virtual reality (VR) have been recommended in numerous studies as distraction interventions to alleviate patient anxiety in various clinical situations [33,34,35]. Distraction techniques encourage the patient to focus on stimuli other than physical sensations [36]. It can relieve anxiety and other physical symptoms such as pain, nausea, and stress from chemotherapy. Patients had an altered perception of time when using VR, which validates the distracting capacity of the intervention [37].

Researchers investigated two virtual environments that simulated nature scenes, including a blue sky and a green field that participants could explore. During the tour, a voice instructed the user and told stories to evoke specific emotions, significantly decreasing negative mood scores and increasing positive mood after four brief 30-min sessions [38]. Our findings demonstrated that using nature images for just 12 min in a single session is sufficient to modulate negative affect, making it clinically relevant. Furthermore, the presented approach offers cost-effective benefits due to its minimal technological resource requirements.

Nature images offer more than just a distraction technique; they engage mechanisms related to aesthetic appreciation. Aesthetic experiences keep the observers focused on the moment, making room for perceived sensations and emotions provoked by the beauty of what is being observed [39]. This aesthetic presence allows viewers to direct attention to perceptual activity for its own sake, which results in an amplified sensory gain. Exposure to the beauty of nature can increase the frequency and intensity of aesthetic experiences. This contributes to the improvement of emotional capacities and greater satisfaction with life. It activates the brain’s reward system and dopaminergic activity, which induces feelings of pleasure and enhances motivation [39]. They are visually (or multisensorily) pleasurable and can help reduce stress. These experiences trigger positive emotions, maintain the state of non-vigilant attention, decrease negative thoughts, and allow the return of physiological excitement to more moderate levels, as advocated by the Psychophysiological Stress Recovery Theory [40].

The implication of the presented study for the field of care assistance is the potential to offer patients a low-cost resource that provides both pleasure, through nature-related well-being images, and the ability to increase positive affect and reduce negative affect during chemotherapy. Furthermore, within the education field, it provides an opportunity to foster discussions among healthcare professionals in training about novel care practices characterized by lightweight technologies, such as the one presented in this study.

One possible limitation of the findings is that the study was conducted in a single oncology service of a private hospital. Additionally, during data collection the number of refusals to participate was considerable, and many patients reported that this was the moment they had to rest or spend time talking to their companion, or that they preferred to pay attention to the medication infusion process.

Future research should focus on applying the proposed intervention in other organizational contexts. The aim is to assess its effectiveness in different patient settings, encompassing various procedures, health facilities, and populations.

## 5. Conclusions

The videos called Beauty, Emotions Up, and Miscellany, composed of positive emotion-inducing nature images from the e-Nature Positive Emotion Photography Database (e-NatPOEM) [16], showed therapeutic potential to reduce the negative affect and clinical symptoms of chemotherapy patients.

The Tranquility video unexpectedly showed no influence on the evaluated clinical and psychological outcomes.

## Figures and Tables

**Figure 1 ijerph-20-06555-f001:**
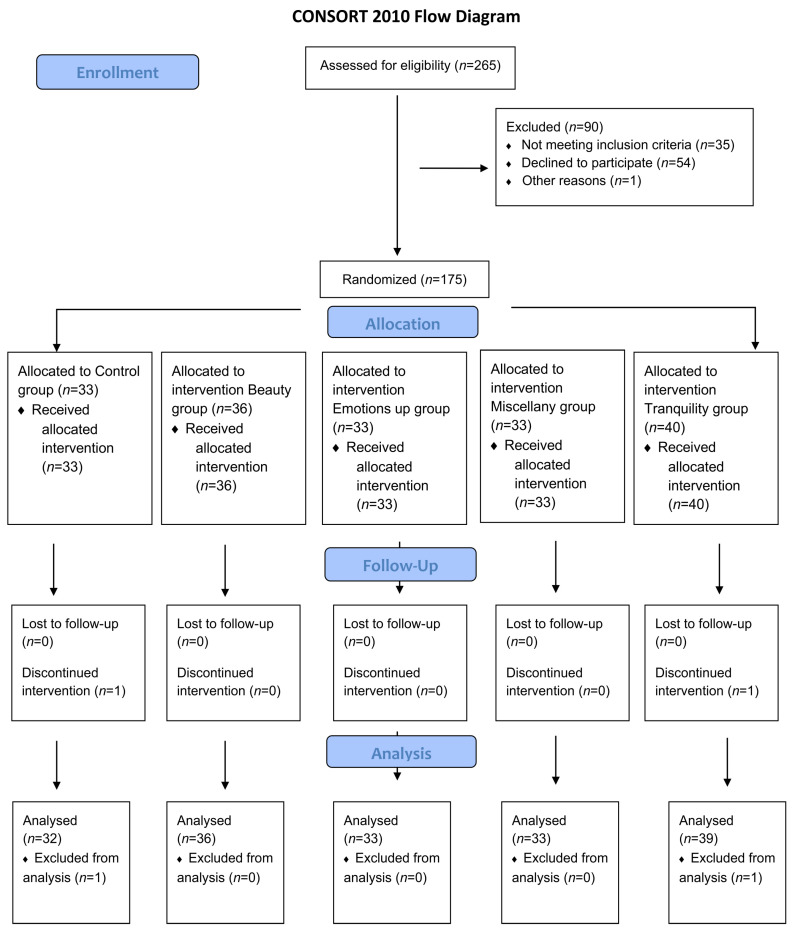
Study flowchart of included and excluded patients.

**Figure 2 ijerph-20-06555-f002:**
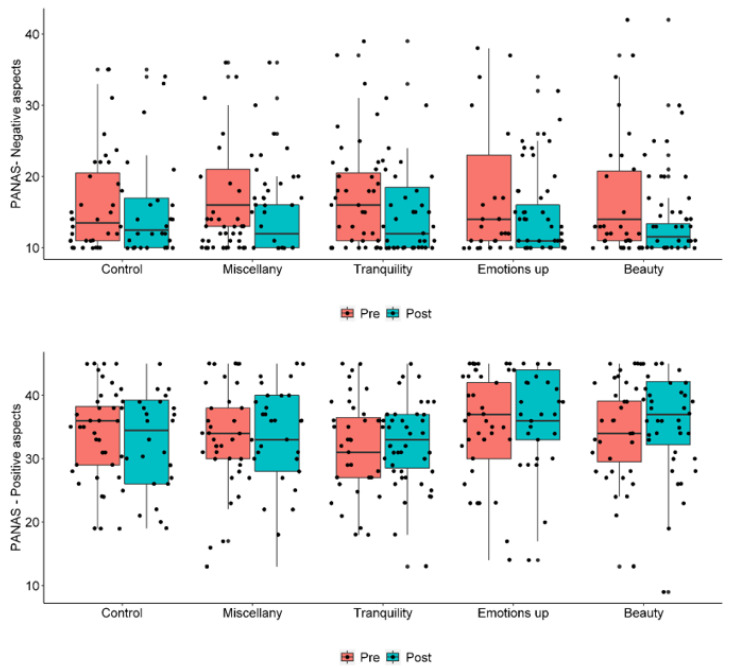
PANAS negative and positive aspects, respectively, by groups and moments.

**Table 1 ijerph-20-06555-t001:** Participants’ demographic characteristics.

	Total	Control	Miscellany	Tranquility	Emotions Up	Beauty
	173	32	33	39	33	36
Age (years):						
mean (standard deviation)	57 (14)	56 (14)	56 (16)	57 (12)	61 (14)	56 (16)
minimum—maximum	22–90	33–86	30–87	23–78	33–79	22–90
Sex—*n* (%)						
Female	98 (56.6)	18 (56.3)	22 (66.7)	22 (56.4)	17 (51.5)	19 (52.8)
Male	75 (43.4)	14 (43.8)	11 (33.3)	17 (43.6)	16 (48.5)	17 (47.2)
Nationality—*n* (%)						
Brasil	162 (93.6)	27 (84.4)	32 (97.0)	39 (100.0)	33(100.0)	31 (86.1)
Others *	11 (6.6)	5 (15.5)	1 (3.0)	0 (0.0)	0 (0.0)	5 (15.2)
Marital status—*n* (%)						
Single	17 (9.8)	3 (9.4)	6 (18.2)	1 (2.6)	3 (9.1)	4 (11.1)
Married/with partner	127 (73.4)	22 (68.8)	23 (69.7)	31 (79.5)	23 (69.7)	28 (77.8)
Divorced	24 (13.9)	7 (21.9)	4 (12.1)	6 (15.4)	5 (15.2)	2 (5.6)
Widowed	5 (2.9)	0 (0.0)	0 (0.0)	1 (2.6)	2 (6.1)	2 (5.6)
Education—*n* (%)						
Primary school	3 (1.8)	2 (6.2)	0 (0.0)	0 (0.0)	0 (0.0)	1 (2.8)
High school	18 (10.4)	2 (6.3)	4 (12.1)	4 (10.3)	2 (6.1)	6 (16.7)
Graduation	92 (53.2)	17 (53.1)	14 (42.4)	23 (59.0)	19 (57.6)	19 (52.8)
Postgraduate	60 (34.7)	11 (34.4)	15 (45.5)	12 (30.8)	12 (36.3)	10 (27.8)
Born in—*n* (%)						
Rural areas	10 (5.8)	4 (12.5)	0 (0.0)	3 (7.7)	3 (9.1)	0 (0.0)
Urban areas	163 (94.2)	28 (87.5)	33 (100.0)	36 (92.3)	30 (90.9)	36(100.0)
Background living area—*n* (%)						
Urban	146 (84.4)	26 (81.3)	31 (93.9)	33 (84.6)	27 (81.8)	29 (80.6)
Both	25 (14.5)	5 (15.6)	2 (6.1)	5 (12.8)	6 (18.2)	7 (19.4)
Rural	2 (1.2)	1 (3.1)	0 (0.0)	1 (2.6)	0 (0.0)	0 (0.0)
Frequency of intentional contact with nature—*n* (%)						
Daily	71 (41.0)	17 (53.1)	13 (39.4)	12 (30.8)	13 (39.4)	16 (44.4)
5–6 days/week	2 (1.2)	0 (0.0)	0 (0.0)	2 (5.1)	0 (0.0)	0 (0.0)
3–4 days/week	14 (8.1)	1 (3.1)	5 (15.2)	5 (12.8)	1 (3.0)	2 (5.6)
1–2 days/week	35 (20.2)	4 (12.5)	7 (21.2)	8 (20.5)	8 (24.2)	8 (22.2)
Fortnightly	16 (9.2)	4 (12.5)	3 (9.1)	4 (10.3)	3 (9.1)	2 (5.6)
Monthly	24 (13.9)	6 (18.8)	4 (12.1)	5 (12.8)	5 (15.2)	4 (11.1)
Yearly	11 (6.4)	0 (0.0)	1 (3.0)	3 (7.7)	3 (9.1)	4 (11.1)

* Others: Bolivia (2), Argentina (1), Spain (1), United States (1), Netherlands (1), Panama (1), Paraguay (1), Peru (1), Portugal (1) and Sweden (1).

**Table 2 ijerph-20-06555-t002:** Characteristics of chemotherapy treatment.

	Total	Control	Miscellany	Tranquility	Emotions Up	Beauty
	173	32	33	39	33	36
1st chemotherapy session in life—*n* (%)						
No	102 (59.0)	18 (56.3)	20 (60.6)	26 (66.7)	20 (60.6)	18 (50.0)
Yes	71 (41.0)	14 (43.8)	13 (39.4)	13 (33.3)	13 (39.4)	18 (50.0)
Time since diagnosis—*n* (%)						
Less than 1 month	15 (8.7)	4 (12.5)	0 (0.0)	4 (10.3)	3 (9.1)	4 (11.1)
1–2 months	34 (19.7)	7 (21.9)	6 (18.2)	8 (20.5)	6 (18.2)	7 (19.4)
3–6 months	44 (25.4)	5 (15.6)	10 (30.3)	8 (20.5)	10 (30.3)	11 (30.6)
6 months-1 year	21 (12.1)	3 (9.4)	5 (15.2)	5 (12.8)	4 (12.1)	4 (11.1)
1–5 years	44 (25.4)	10 (31.3)	8 (24.2)	9 (23.1)	9 (27.3)	8 (22.2)
More than 5 years	15 (8.7)	3 (9.4)	4 (12.1)	5 (12.8)	1 (3.0)	2 (5.6)
Estimated duration of treatment—*n* (%)						
Less than 1 month	3 (1.7)	1 (3.1)	2 (6.1)	0 (0.0)	0 (0.0)	0 (0.0)
1–2 months	11 (6.4)	1 (3.1)	0 (0.0)	4 (10.3)	2 (6.1)	4 (11.1)
3–6 months	44 (25.4)	11 (34.4)	7 (21.2)	9 (23.1)	7 (21.2)	10 (27.8)
6 months-1 year	52 (30.1)	8 (25.0)	11 (33.3)	9 (23.1)	12 (36.4)	12 (33.3)
1–5 years	43 (24.9)	7 (21.9)	9 (27.3)	11 (28.2)	9 (27.3)	7 (19.4)
More than 5 years	20 (11.6)	4 (12.5)	4 (12.1)	6 (15.4)	3 (9.1)	3 (8.3)

**Table 3 ijerph-20-06555-t003:** PANAS negative and positive aspects comparisons between pre- and post-intervention moments and groups.

	Pre-Intervention	Post-Intervention	*p*-Value Pre vs. Post Intervention *
	Estimated Mean (CI 95%)	Estimated Mean (CI 95%)	
Negative aspects			
Intervention	17.15 (16.01; 18.38)	14.45 (13.42; 15.56)	**<0.001**
Control	16.30 (13.95; 19.06)	14.96 (12.89; 17.36)	**0.034**
*p*-value (Groups)	0.553	0.688	
Negative aspects			
Beauty	16.86 (14.60; 19.47)	13.90 (11.96; 16.17)	**0.017**
Emotions Up	17.24 (14.81; 20.08)	14.52 (12.48; 16.88)	**0.013**
Tranquility	16.97 (14.96; 19.24)	14.98 (13.02; 17.23)	**0.038**
Miscellany	17.61 (15.45; 20.07)	14.36 (12.39; 16.65)	**<0.001**
Control	16.30 (13.95; 19.06)	14.96 (12.89; 17.36)	**0.034**
*p*-value (Groups)	0.963	0.953	
Positive aspects			
Intervention	33.88 (32.76; 35.05)	34.16 (32.80; 35.58)	0.620
Control	34.01 (31.76; 36.43)	32.92 (30.25; 35.82)	0.189
*p*-value (Groups)	0.924	0.433	
Positive aspects			
Beauty	34.00 (31.87; 36.27)	35.49 (32.81; 38.39)	0.268
Emotions Up	35.94 (33.61; 38.43)	35.61 (32.70; 38.77)	0.763
Tranquility	31.89 (29.75; 34.19)	32.46 (30.32; 34.75)	0.458
Miscellany	34.06 (32.04; 36.21)	33.27 (30.40; 36.42)	0.499
Control	34.01 (31.76; 36.43)	32.92 (30.25; 35.82)	0.189
*p*-value (Groups)	0.203	0.322	

* Significant *p*-values (<0.05) are identified in bold.

**Table 4 ijerph-20-06555-t004:** Symptoms evaluated by the ESAS: comparisons between pre- and post-intervention moments and groups.

	Pre-Intervention	Post-Intervention	*p*-Value Pre vs. Post Intervention *
	Estimated Mean (CI 95%) *	Estimated Mean (CI 95%)	
Pain			
Intervention	0.97 (0.72; 1.31)	0.66 (0.46; 0.95)	**<0.001**
Control	1.13 (0.76; 1.68)	0.88 (0.50; 1.53)	0.258
*p*-value (Groups)	0.574	0.433	
Tiredness			
Intervention	2.82 (2.41; 3.29)	2.26 (1.88; 2.73)	**0.002**
Control	3.59 (2.71; 4.77)	3.03 (2.19; 4.20)	0.086
*p*-value (Groups)	0.169	0.160	
Nausea			
Intervention	0.75 (0.52; 1.08)	0.60 (0.40; 0.90)	0.157
Control	1.41 (0.80; 2.48)	1.13 (0.58; 2.20)	0.297
*p*-value (Groups)	0.128	0.196	
Depression			
Intervention	1.90 (1.51; 2.40)	1.32 (1.00; 1.74)	**0.003**
Control	1.91 (1.21; 3.01)	1.34 (0.84; 2.15)	0.139
*p*-value (Groups)	0.991	0.947	
Anxiety,			
Intervention	3.01 (2.56; 3.55)	2.12 (1.72; 2.62)	**<0.001**
Control	3.06 (2.23; 4.20)	2.41 (1.68; 3.45)	0.046
*p*-value (Groups)	0.930	0.566	
Drowsiness			
Intervention	2.88 (2.46; 3.38)	2.96 (2.53; 3.46)	0.738
Control	3.47 (2.55; 4.71)	3.50 (2.60; 4.72)	0.931
*p*-value (Groups)	0.319	0.352	
Appetite			
Intervention	3.45 (2.99; 3.97)	2.72 (2.30; 3.23)	**0.001**
Control	3.75 (2.80; 5.02)	3.22 (2.32; 4.47)	0.220
*p*-value (Groups)	0.619	0.400	
Wellbeing			
Intervention	3.30 (2.90; 3.75)	2.91 (2.50; 3.38)	0.054
Control	4.13 (3.22; 5.28)	3.63 (2.74; 4.80)	0.365
*p*-value (Groups)	0.142	0.204	
Shortness of breath			
Intervention	0.74 (0.48; 1.14)	0.52 (0.32; 0.84)	0.050
Control	0.59 (0.21; 1.71)	0.41 (0.13; 1.23)	0.579
*p*-value (Groups)	0.690	0.671	
Sleep disturbance			
Intervention	2.86 (2.46; 3.32)	2.80 (2.40; 3.27)	0.753
Control	4.03 (3.14; 5.18)	3.41 (2.51; 4.62)	0.171
*p*-value (Groups)	**0.036**	0.291	

* Significant *p*-values (<0.05) are identified in bold.

**Table 5 ijerph-20-06555-t005:** Symptoms evaluated by the ESAS: comparisons between moments and groups, specifying interventions—1st part.

	Pre-Intervention	Post-Intervention	*p*-Value Pre vs. Post Intervention *
	Estimated Mean (CI 95%)	Estimated Mean (CI 95%)	
Pain			
Beauty	0.69 (0.35; 1.37)	0.38 (0.11; 1.26)	**0.001**
Emotions Up	1.15 (0.67; 1.99)	0.76 (0.37; 1.57)	**0.035**
Tranquility	1.18 (0.67; 2.08)	0.79 (0.46; 1.36)	0.063
Miscellany	0.85 (0.47; 1.53)	0.70 (0.35; 1.39)	0.393
Control	1.13 (0.76; 1.68)	0.88 (0.50; 1.53)	0.258
*p*-value (Groups)	0.612	0.616	
Tiredness			
Beauty	1.92 (1.24; 2.95)	1.92 (1.29; 2.85)	>0.99
Emotions Up	3.12 (2.34; 4.16)	2.15 (1.44; 3.22)	**0.003**
Tranquility	2.95 (2.28; 3.81)	2.82 (2.06; 3.86)	0.713
Miscellany	3.33 (2.47; 4.49)	2.09 (1.42; 3.09)	**<0.001**
Control	3.59 (2.71; 4.77)	3.03 (2.19; 4.20)	0.086
*p*-value (Groups)	0.085	0.315	
Nausea			
Beauty	0.42 (0.20; 0.88)	0.36 (0.14; 0.96)	0.682
Emotions Up	0.94 (0.45; 1.96)	0.58 (0.23; 1.42)	0.057
Tranquility	0.72 (0.35; 1.49)	0.87 (0.44; 1.72)	0.543
Miscellany	0.97 (0.53; 1.77)	0.58 (0.29; 1.14)	0.071
Control	1.41 (0.80; 2.48)	1.13 (0.58; 2.20)	0.297
*p*-value (Groups)	0.115	0.353	
Depression			
Beauty	2.08 (1.31; 3.32)	0.94 (0.45; 1.97)	**0.031**
Emotions Up	1.61 (0.87; 2.96)	1.33 (0.78; 2.28)	0.328
Tranquility	1.85 (1.22; 2.79)	1.87 (1.22; 2.88)	0.941
Miscellany	2.06 (1.41; 3.02)	1.06 (0.60; 1.89)	**0.002**
Control	1.91 (1.21; 3.01)	1.34 (0.84; 2.15)	0.139
*p*-value (Groups)	0.956	0.473	
Anxiety			
Beauty	2.94 (2.08; 4.17)	1.61 (0.95; 2.74)	**0.006**
Emotions Up	2.85 (1.99; 4.08)	2.06 (1.37; 3.09)	**0.043**
Tranquility	2.85 (2.08; 3.89)	2.54 (1.84; 3.51)	0.459
Miscellany	3.45 (2.61; 4.58)	2.24 (1.41; 3.57)	**0.003**
Control	3.06 (2.23; 4.20)	2.41 (1.68; 3.45)	**0.046**
*p*-value (Groups)	0.902	0.596	

* Significant *p*-values (<0.05) are identified in bold.

**Table 6 ijerph-20-06555-t006:** Symptoms evaluated by the ESAS: comparisons between moments and groups, specifying interventions—2nd part.

	Pre-Intervention	Post-Intervention	*p*-Value Pre vs. Post Intervention *
	Estimated Mean (CI 95%) *	Estimated Mean (CI 95%)	
Drownsiness			
Beauty	2.78 (1.98; 3.89)	2.83 (2.07; 3.88)	0.883
Emotions Up	2.48 (1.73; 3.58)	2.58 (1.74; 3.80)	0.807
Tranquility	3.08 (2.38; 3.98)	3.33 (2.55; 4.36)	0.613
Miscellany	3.15 (2.27; 4.38)	3.03 (2.23; 4.12)	0.831
Control	3.47 (2.55; 4.71)	3.50 (2.60; 4.72)	0.931
*p*-value (Groups)	0.681	0.702	
Appetite			
Beauty	2.83 (2.02; 3.98)	2.42 (1.62; 3.60)	0.231
Emotions Up	3.58 (2.74; 4.67)	2.33 (1.60; 3.40)	**0.006**
Tranquility	4.15 (3.31; 5.22)	3.56 (2.73; 4.65)	0.142
Miscellany	3.15 (2.32; 4.29)	2.45 (1.75; 3.44)	0.158
Control	3.75 (2.80; 5.02)	3.22 (2.32; 4.47)	0.220
*p*-value (Groups)	0.354	0.248	
Wellbeing			
Beauty	3.11 (2.35; 4.12)	2.22 (1.58; 3.13)	**0.024**
Emotions Up	3.76 (2.94; 4.81)	3.30 (2.47; 4.42)	0.233
Tranquility	3.38 (2.70; 4.25)	3.26 (2.54; 4.18)	0.758
Miscellany	2.94 (2.27; 3.81)	2.85 (2.06; 3.94)	0.821
Control	4.13 (3.22; 5.28)	3.63 (2.74; 4.80)	0.365
*p*-value (Groups)	0.364	0.176	
Shortness of breath			
Beauty	0.50 (0.21; 1.17)	0.22 (0.06; 0.78)	**0.038**
Emotions Up	1.30 (0.64; 2.65)	1.09 (0.53; 2.25)	0.432
Tranquility	0.54 (0.21; 1.35)	0.49 (0.19; 1.24)	0.793
Miscellany	0.67 (0.25; 1.81)	0.30 (0.11; 0.87)	0.200
Control	0.59 (0.21; 1.71)	0.41 (0.13; 1.23)	0.579
*p*-value (Groups)	0.641	0.321	
Sleep disturbance			
Beauty	2.28 (1.63; 3.18)	2.83 (2.04; 3.93)	0.108
Emotions Up	2.48 (1.78; 3.46)	2.33 (1.62; 3.37)	0.618
Tranquility	3.38 (2.57; 4.45)	3.10 (2.39; 4.03)	0.342
Miscellany	3.24 (2.53; 4.16)	2.88 (2.14; 3.86)	0.436
Control	4.03 (3.14; 5.18)	3.41 (2.51; 4.62)	0.171
*p*-value (Groups)	**0.040**	0.581	

* Significant *p*-values (<0.05) are identified in bold.

**Table 7 ijerph-20-06555-t007:** Summary of clinical improvement observed in each group.

Group	Clinical Improvement
Beauty	Pain, Depression, Anxiety, Well-being, Shortness of breath
Emotions Up	Tiredness, Depression, Anxiety
Tranquility	Pain, Tiredness, Anxiety, Appetite
Miscellany	No significant improvement
Control	Anxiety

## Data Availability

The datasets generated during and/or analyzed during the current study are available upon request from the corresponding author.

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
