# Peer review of "Nature Photographs as Complementary Care in Chemotherapy: A Randomized Clinical Trial"

_ijerph, 2023, doi:10.3390/ijerph20166555_

Round 1
Reviewer 1 Report
Review: Nature photographs as complimentary care in chemotherapy: a randomized clinical trial
The study investigated the effect that viewing photographs from nature has on the physiological and psychological wellbeing of patients with cancer during treatment. The sample is large and consists of 4 groups including a control group. The study makes use of three psychometric instruments of which two the PANAS and the Edmonton Symptom Assessment System, are used during pre and post test.
The study is well executed and written
Abstract
The abstract is well-structured and provides relevant information about the study, including the objective, method, results, and conclusions. The use of nature-based interventions for cancer patients undergoing chemotherapy is a promising area of research, and this study contributes to the growing body of evidence supporting the use of nature images as a tool to help control clinical and psychological symptoms in cancer patients. Overall, the abstract is clear, informative, and well-written.
References
· The resources are current and conforms to journal style.
Strengths of the paper
· Important and novel topic
· Large sample
· Rigorous methodology
· Fascinating and meaningful results
· Discussion well integrated with Lit Review and Results
Weakness of the paper
· None
Specific issues
· P.2, 4th paragraph: Citation typing error (Ulrich, 1984)
· P.6, beneath Bullet 4. – seems like a repetition
· P.13, last para – please explain what ISO is? I could not find an explanation/definition.
· Conclusion:
o Please include the limitations of the study,
o Future research
o Implication of the study
Well done on an excellent study!
Author Response
Dear Reviewer 1,
The authors would like to thank you for your consideration and for the valuable comments, that effectively contributed to the improvement of the paper.
In the revised paper, among other aspects, we focused on presenting a careful revision of written English, complementary and new explanations of various points (including new references, highlighted in yellow).
All green highlights in the revised manuscript correspond to improvements in the English language, such as: grammar, sentence structure, vocabulary, clarity, and overall linguistic accuracy. It is important to note that any text not highlighted in green, unless explicitly mentioned in this letter, represents new additions incorporated in response to feedback from other reviewers.
We would like to mention that all the comments of the reviewer were carefully addressed in the following point-by-point responses, and we hope that the outcomes fulfill the Reviewer’s expectations.
Yours sincerely,
The authors

Reviewer 2 Report
In this article, the authors describe the utilization of a set of nature photographs with different topics during chemotherapy of cancer patients. They applied a randomization into four groups with different topics of photographs, namely Tranquility, Beauty, Emotions-up, Miscellany, and a control group. They find that negative emotions, anxiety, depression and cancer-related symptoms are reduced to different extent in these groups of patients.
The study is relevant since it may be an easily feasible intervention, from which cancer patients may benefit during chemotherapy. It is a addition to the methods of supportive therapy during chemotherapy. There are, however, several points of concern, that should be addressed by the authors.
1. A major concern towards the significance of the presented results is related to the absence of any information regarding the medication used in these patients. In particular, it is not clear from the presentation whether the proportion of highly vs. moderately emetogenic chemotherapy was balanced in the different intervention and experimental groups.
2. In addition, the antiemetic medication may largely influence the patients' psychological state since dexamethasone, neuroleptic or antidepressant drugs might be used, especially in highly emetogenic chemotherapy regimes. Furthermore, some drugs used for pain control may exert psychological side effects, e.g., morphines and tramadolol. It is recommended to provide information on the use of concommitant medications in the five groups, and whether there were major imbalances.
3. In the Materials and Methods section, the process of screening and randomization should be described more in detail: Were the 265 screened patients consecutive cancer patients in a certain time period, or was inclusion to the study at the dicretion of the investigators? Over which period of time did the randomization take place?
4. Did the controls receive any special attention or materials? From Figure 2 and Table 3, they experienced significant reduction of negative feelings (p=0.032), and Table 4 shows a significant reduction of anxiety (p=0.046) in the pre-post comparison. These observations need to be discussed since they influence the interpretation of the reported improvements in the intervention groups.
5. The use of nature photographs may be related to distraction techniques of psychotherapy. The author have mentioned this briefly (p.14), but the paper would certainly gain from deepening the discussion with respect to the underlying psychological theory. There are several reports on distraction as a means of reducing anxiety in children and adults, including visual arts or virtual reality.
6. Despite the previous description of the e-NatPOEM database of photographs by the investigatos (reference 14), it should be cautioned in the Discussion that this is no validated instrument of psychology.
Minor: There are some spelling errors that should be corrected, e.g., "afectts".
Quality of English language is generally o.k. Some expressions may be mproved. A few spelling errors should be corrected.
Author Response
Dear Reviewer 2,
The authors would like to thank you for your consideration and for the valuable comments that effectively contributed to the improvement of the paper.
In the revised paper, among other aspects, we focused on presenting a careful revision of written English, complementary and new explanations of various points (including new references, highlighted in yellow).
All green highlights in the revised manuscript correspond to improvements in the English language, such as: grammar, sentence structure, vocabulary, clarity, and overall linguistic accuracy. It is important to note that any text not highlighted in green, unless explicitly mentioned in this letter, represents new additions incorporated in response to feedback from other reviewers.
We would like to mention that all the comments of the reviewer were carefully addressed in the following point-by-point responses, and we hope that the outcomes fulfill the Reviewer’s expectations.
Yours sincerely,
The authors

Reviewer 3 Report
I believe this paper addresses something that is important. The introduction is grounded in history (Florence Nightingale's experience and writings) and the study is very innovative and well planned. They also cite literature that has shown the importance of "direct contact with nature" in patient well-being. The authors recruited an outstanding number of patients (173) and used good measurement tools (PANAS, ESAS).
However, the discussion section is very difficult to interpret and understand. Extensive English editing is needed. The third paragraph in this section does not make sense to this reviewer. It is unclear what point the authors are attempting to make comparing their study participants to individuals in the general population or with advanced cancer. Also, small details such as "affects" (does not need an "s") and contents (also does not need an "s") detract from the readability of the article.
One example of an extremely long and confusing sentence is: "These different images when presented together may have had an impact on a sensation closer to the reality observed in nature from a broader perspective (not just the repetition of one or two natural elements), such as working ants, butterflies, paths with flowers, landscapes, which may favor a feeling of proximity and familiarity, as well as presenting more attractive colors that confer vivacity to the imagines that may have influenced the perception of fatigue."
Poor English grammar and form.
Author Response
Dear Reviewer 3,
The authors would like to thank you for your consideration and for the valuable comments that effectively contributed to the improvement of the paper.
In the revised paper, among other aspects, we focused on presenting a careful revision of written English, complementary and new explanations of various points (including new references, highlighted in yellow).
All green highlights in the revised manuscript correspond to improvements in the English language, such as: grammar, sentence structure, vocabulary, clarity, and overall linguistic accuracy. It is important to note that any text not highlighted in green, unless explicitly mentioned in this letter, represents new additions incorporated in response to feedback from other reviewers.
We would like to mention that all the comments of the reviewer were carefully addressed in the following point-by-point responses, and we hope that the outcomes fulfill the Reviewer’s expectations.
Yours sincerely,
The authors

Round 2
Reviewer 2 Report
The authors have markedly imroved their paper. I have no relevant concerns.
Reviewer 3 Report
The authors have made good editorial changes to the document that add to the readability and grammar. They have attempted to address the potential limitations of the study (i.e. high number of subject refusals to participate and the fact that simple interaction with study staff could be responsible for the positive affect.) The study seems acceptable for publication in its current iteration.